# Effectiveness and Safety of Attic Exposition–Antrum Exclusion versus Canal Wall-Up in Patients with Acquired Stage Ib and II Cholesteatoma Affecting the Attic and Tympanic Cavity

**DOI:** 10.3390/jcm12010049

**Published:** 2022-12-21

**Authors:** Francisco Arias Marzán, Esteban Reinaldo Pacheco Coronel, Ayoze Lemes Robayna, Maria Cecilia Salom Lucena, Gemma De Lucas Carmona, María Gabriela Muñoz Cordero, Diego Hernando Macias Rodríguez, Alejandro Jimenez Sosa

**Affiliations:** 1Otorhinolaryngology Service, Hospital Universitario de Canarias, Ofra s/n, 38320 La Laguna, Tenerife, Spain; 2Research Unit, Hospital Universitario de Canarias, Ofra s/n, 38320 La Laguna, Tenerife, Spain

**Keywords:** cholesteatoma, attic exposition–antrum exclusion, wall-up tympanomastoidectomy

## Abstract

This study aims to test the effectiveness and safety of exteriorization surgery comprising atticotomy and obliteration of the additus ad antrum, also referred to as attic exposition–antrum exclusion (AE-AE) surgery. This surgery combines otoendoscopy with surgical microscopy for the treatment of acquired pars flaccida cholesteatoma in stages Ib and II (according to the classification of the Japan Otological Society) present in the attic and the tympanic cavity. We reviewed a historical cohort of 65 patients. Of the total, 21 were treated with canal wall-up tympanomastoidectomy (CWU). Patients in whom the AE-AE technique was performed had residual and recurrence rates of 0% and 9.1%, respectively, compared with 28.6% and 9.5%, respectively, for those treated with CWU. In the AE-AE procedure, surgery is performed in one stage compared with the two stages in CWU, to address the risk of residual cholesteatoma. Auditory thresholds were higher in the CWU group compared with the AE-AE group in the pre-surgery (53 ± 16 vs. 44 ± 15 dB; *p* = 0.039) and post-surgery (52 ± 18 vs. 42 ± 16 dB; *p* = 0.042) evaluations but not in pre–post-surgery comparisons for either the AE-AE technique (*p* = 0.89) or the CWU technique (*p* = 0.96). We conclude that AE-AE is an effective and safe technique for the treatment of acquired stage Ib and II cholesteatoma present in the attic and tympanic cavities.

## 1. Introduction

Middle-ear cholesteatoma is characterized by the existence of epithelial tissue with aggressive and destructive behavior in the tympanic and mastoid cavities [1]. Although its prevalence differs according to ethnic groups, the annual incidence of acquired cholesteatoma is estimated at between 9 and 12.6 cases/100,000 adults and 3 and 15 cases/100,000 children [2]. As an alternative, Olaizola et al. [3] described an on-demand technique for the treatment of cholesteatoma called attic exposition–antrum exclusion (AE-AE).

AE-AE is technically and conceptually characterized by the complete exposure of the attic by reaming the upper wall of the external auditory canal (in a procedure termed a transcanal epitympanectomy), which allows the exclusion of the antrum and mastoid via closure of the additus through cartilaginous grafts. Using the classification of the IOOG, the categorization of this technique corresponds to an atticotomy with partial obliteration of the additus [4]. The attic is then successfully exposed to allow complete excision of the cholesteatoma, resulting in reduced probabilities of residual disease and recurrence of acquired cholesteatoma, which frequently affect this space. The closure of the mastoid space reduces the exposed space and promotes self-cleaning of the attic, allowing the entry of water without a vertigo crisis and with a low risk of infection. This technique does not interfere with the performance of ossiculoplasty.

Finally, we find that the use of otoendoscopy in this technique reduces recurrence due to persistent cholesteatoma, mainly in the epitympanum and mastoid antrum. The introduction of an endoscope is also significant in ossicular reconstruction.

The objective of this study is to estimate the effectiveness and safety of AE-AE with endoscopic support when used to treat cholesteatoma occupying the attic and tympanic cavity in stages Ib and II according to the classification of the Japan Otological Society [1] (Table 1 and Figure 1).

## 2. Materials and Methods

### 2.1. Design and Patients

Using a historical cohort design, 65 patients undergoing their first surgeries between 2001 and 2020 with a diagnosis of stage Ib or II acquired attic and tympanic cholesteatoma were included [1]. CWU procedures were performed for the first 12 years, and we changed to the AE-AE procedure in the last 7 years (Figure 2 and Table 2).

This research was approved by the Ethics Committee (2020/86). All patients were operated on with either (a) the AE-AE technique with the use of an otoendoscope or (b) the canal wall-up tympanomastoidectomy technique with only a microscope. The stage of cholesteatoma was established from the operative findings recorded in the clinical history as well as by reviewing the extent described in the surgical protocol and confirming the histopathological diagnosis of the pathological anatomy.

All patients underwent preoperative computed axial tomography and surgery with intraoperative facial nerve monitoring according to a previously published protocol [5]. Audiometry was performed before surgery, as well as post-surgery after 1 or more years, after the first surgery for AE-AE and after the second procedure for CWU. We measured the hearing levels of the bone conduction (BC) and air conduction (AC) in the 500, 1000, and 2000 Hz frequencies (pure tone average: PTA) to assess hearing levels and hearing gain [5]. In addition, we compared PTA and high-frequency (>2000 Hz) deterioration.

We determined the difference between the AC and BC PTAs (ABG) pre- and post-surgery and determined whether the values were less than 20 dB, which was used as the criterion for a good post-surgical hearing outcome. When necessary, patients underwent an ossiculoplasty procedure in the same surgical act as cholesteatoma exeresis, using either pinna or tragus cartilage over the stapes or total or partial titanium prosthesis. Treatment of the tympanic membrane was performed with temporalis muscle fascia or tragus perichondrium.

The inclusion criteria were as follows: attic cholesteatoma of stage Ib or II affecting only the attic or both the attic and the tympanic cavity; first intervention on the pathological ear; and patients undergoing AE-AE with the use of a microscope and an otoendoscope or undergoing CWU with only the use of a microscope. The exclusion criteria were the absence of cholesteatoma during surgery, cholesteatoma not acquired from pars flaccida, and prior surgery in the affected ear. 

Age, sex, affected ear, residual disease, and time of recurrence (if any) were also recorded in the clinical history.

### 2.2. Surgical Procedure

Five surgeons with more than 10 years of experience in otologic surgery operated on the patients using the CWU technique. These surgeons were not exclusively dedicated to the treatment of ear pathologies but were specialists in otolaryngology and general cervical pathology. The series of patients who underwent CWU were operated on between 2001 and 2013, a period prior to the creation of super-specialty units in our hospital. The otology unit currently has 2 surgeons who have performed all surgeries with microscopes and endoscopes since 2013. We performed AE-AE hybrid surgeries using standard microsurgical and otoendoscopy instruments.

The following is a description of the surgical steps, which are based on the IOOG categorization [4]. 

CWU reconstruction technique:Retroauricular approach (A4);Tympanomeatal flap is lifted, and the middle ear is explored;Excision of middle-ear cholesteatoma and exploration of ossicular chain and incudostapedial articulation;Mastoidectomy with canal wall preserved and posterior tympanotomy M1b with additional combination of M1b and M2a (atticotomy);No external ear canal reconstruction (Ex);No obliteration (Ox);No bone removal from external ear canal wall (Ax);Tympano-ossicular reconstruction if necessary (Tx, Tn, T1, and T2; Ox, On, and Ost).

All cases were scheduled for second-look (S2p) revision between 1 and 2 years. Surgeons did not use otoendoscopes or diffusion MRI to assess residual disease.

AE-AE hybrid exteriorization technique is shown in Figure 3.

Following the steps previously described by other authors [6,7,8], we added the use of an otoendoscope 14 mm in length and 2.7 mm in diameter with angulations of 0 and 30°.

9.Retroauricular approach (A4);10.Tympanomeatal flap is lifted, and middle ear is explored;11.Excision of middle-ear cholesteatoma and exploration of ossicular chain and incudostapedial articulation using endoscope and microscope;12.Atticotomy (M2a). Superior wall of EAC is drilled with two hands and microscopic control, exposing entire tegmen tympani. Once attic is completely exposed (Figure 4), cholesteatoma is removed. If it invades medial region of attic, it is necessary to remove incus and head of malleus. At this point, we use an otoendoscope at 0 and 30° degrees to thoroughly check the epitympanum, sinus tympani, and antrum;13.No external ear canal reconstruction (Ex);14.Partial obliteration of additus (O1) using cartilage. Once cholesteatoma is removed, we proceed to closure of additus and, therefore, antroexclusion using several pieces of cartilage. Here, again, the use of an endoscope allows for perfect antroexclusion.15.No bone removal from external ear canal wall (Ax);16.Tympano-ossicular reconstruction, if necessary (Tx, Tn, T1, and T2; Ox, On, and Ost), using endoscope and microscope.17.Superior meatoplasty. To favor marsupialization of attic and self-cleaning, we perform an upper meatoplasty by incision at 12 o’clock in upper wall of EAC and removal of cartilaginous tissues. Place tamponade with gauze edge soaked in antibiotic and corticosteroid creams and remove after 10 days.

Using endoscopes in different stages allowed us to access areas of the middle ear for which visualization was very difficult, allowing cholesteatoma elimination and epithelium cleaning, mainly in the epitympanum and tympanic sinus in addition to the hypotympanum, protympanum, and supratubaric fossa.

#### Limitations 

This study is observational in nature, involving a comparison of the surgical performances of otologists (2 surgeons) and general ENT specialists (5 surgeons) during different periods of time. Although the patients were not randomized, the fact that the recurrence rates are similar to those reported in other studies leads us to conclude that the surgeon is not a factor influencing the recurrence rate. We do not know if the results would change (better outcomes) if the otologists had performed the CWU procedures, although we consider super-specialization to be important in ENT surgeries.

### 2.3. Statistical Analysis

Categorical variables are expressed as frequencies and percentages. Quantitative variables are expressed as means and standard deviations. The proportions between surgical groups for sex, affected ear, stage of cholesteatoma, facial nerve dehiscence, reconstruction, recurrence, and PTA were compared using chi-squared tests or Fisher’s exact tests, when appropriate. Comparisons between surgical groups for age, time to recurrence, and audiometry were carried out using Mann–Whitney tests. The determination of the pre–post-surgery period within each surgical group was carried out with Friedman or Wilcoxon tests, as appropriate. 

All *p*-values below 0.05 were considered to indicate statistically significant differences. Statistical analysis was performed using SPSS (IBM Corp. Released 2017. IBM SPSS Statistics for Windows, Version 25.0. Armonk, NY, USA) and LogXact-4 for Windows (Version 4.1. Cytel Software Corporation. Cambridge, MA, USA).

## 3. Results

Sixty-five patients were included in the analysis. They were divided into two groups according to whether the surgical technique used was AE-AE with endoscopes or CWU. The distribution and demographic variables are shown in Table 2. No differences were observed between the two groups regarding age, sex, or affected ear. The surgery time was shorter in the AE-AE group (146 ± 30.7 min) than in the CWU group (196 ± 57 min), with *p* = 0.005.

Regardless of the type of surgery, the residual disease and recurrence rates were higher in patients with stage II cholesteatoma (11/42; 26.2%) than in those with stage Ib cholesteatoma (1/23; 4.3%), with *p* = 0.04.

The residual disease, but not recurrence, rate was lower for patients in whom the AE-AE technique was performed compared with those treated using CWU (see Figure 5): 28.6% and 9.5%, respectively, in CWU patients and 0% and 9.1%, respectively, in AE-AE patients. Specifically, we found a higher residual disease rate in patients with stage II cholesteatoma of 6/16 (33.33%) for CWU vs. 0/26 (0%) for AE-AE as well as higher recurrence of 2/16 (12.5%) for CWU vs. 3/26 (11.5%) for AE-AE. In stage II cholesteatoma patients, cases of recurrent cholesteatoma were 100% for those treated with CWU vs. 75% for AE-AE.

In the CWU group, 6 (28.6%) patients had an affected sinus tympani due to previous surgery compared with 8 (18.2%) patients in the AE-AE group. In both groups, no recurrence was found for previously affected sinus tympanicum ears. All cases of recurrence in the CWU group were due to failure in attic reconstruction, whereas failure in the AE-AE group was due to an error in the obliteration of the additus ad antrum. 

The most frequent reconstruction technique for the chain was cartilage over stapes, both in the AE-AE group (*n* = 26 (59.1%)) and in the CWU group (*n* = 11 (52.4%)). No difference was found in surgery time between the stage Ib group (158 ± 58 min) and the stage II group (181 ± 51 min), with *p* = 0.17. No differences were observed in the neurophysiological dehiscence of the facial tympanic portions.

Auditory thresholds were higher in the CWU group than in the AE-AE group in the pre-surgery evaluation (53 ± 16 dB vs. 44 ± 15 dB, respectively; *p* = 0.039) but not at one or more years post-surgery (52 ± 52 dB vs. 42 ± 17 dB, respectively; *p* = 0.10) or in pre–post-surgery comparisons, either within the AE-AE technique (*p* = 0.89) or within the CWU technique (*p* = 0.96). Overall, no differences in PTA were found between both groups (*p* = 0.49) or when comparing the hearing levels of partial reconstruction with cartilage over stapes (Table 3).

The continuity of the ossicular chain was restored in 85.7% of the CWU patients and 77.3% of the AE-AE patients. The most frequently used ossiculoplasty method in both groups was the use of cartilage over the stapes suprastructure, whereas reconstruction with total titanium prosthesis was less frequent.

Finally, we found high-frequency losses (>2000 Hz) greater than 20 Db post-surgery in four patients from the CWU group and one patient from the AE-AE group. All of these patients had previous high-frequency affectations prior to surgery. 

## 4. Discussion

The main objective of surgical treatment for cholesteatoma is removal with a minimum rate of recurrence. Canal wall-up and canal wall-down techniques have classically been performed depending on multiple factors. For stage Ib or II cholesteatoma of the attic and tympanic cavity, we modified an AE-AE technique performed in the 1980s [3,6,7,8] in which the drilling and exteriorization of the attic allowed optimal access to the epitympanum (see Figure 6). By adding the use of endoscopes at 0 and 30°, we have improved the control of spaces, such as the anterior epitympanum and sinus tympani, sites where there is frequent cholesteatoma persistence and recurrence, as well as the supratubaric fossa. 

We provide evidence here for lower residual and recurrence rates in acquired stage Ib and II cholesteatoma of the attic and tympanic cavity when using the AE-AE (0–9.1%) technique in combination with otoendoscopy in comparison with the CWU technique (28.6–9.5%). When we contrast our results with those of other authors, we can see that the CWU technique has residual and recurrence rates between 17 and 61% [9]. Other authors [10] reported residual and recurrence rates of 11% to 27% and 5% to 13%, respectively, for patients undergoing closed procedures, whereas residual and recurrence rates are 2% to 10%, respectively, for patients undergoing open procedures. On other hand, the recurrence rates for AE-AE reported in some studies ranges from 4.2% [11] and 4.8% [8] to 8.7% [7]. In all these examples, the reported results are similar to ours.

The impression that the residual rate in the CWU group is high is due to a number of factors. Mainly, exposure of the attic, antrum, facial recess, and sinus timpani is more limited following the CWU procedure, which may lead to difficulty in the complete removal of all involved air cell tracts and in the elimination of cholesteatoma during the initial procedure [12]. Due to the risk of residual and recurrent disease, the closed approach is usually followed by either a staged second look or MRI with diffusion-weighted imaging and, now, the use of endoscopes. In this case, we reviewed the historical cohort and found that the surgeons did not have the possibility of using MRI at the time of the initial surgeries (2001–2013), and they conducted scheduled second-look procedures 1–2 years after the first surgery. 

Second, the epitympanum and mastoid cavity are ultimately relined with nitrogen-absorbing cuboidal mucosal epithelium after the CWU procedure, and the presence of this mucosa can result in the development of continued mucosal inflammation and recurrence [13]. 

We conclude that there are several factors contributing to the decrease in the recurrence rate for the AE-AE procedure. As described in other articles [7,8,11], a significant number of recurrences are observed in superior tympanic retractions; having the attic externalized toward the EAC eliminates this possibility, and having the antrum excluded prevents the emigration of epithelium from the attic toward the mastoid. The fact that the antrum was excluded could suggest that we lost control over possible mastoid recurrence, but this did not occur. 

In contrast to other authors [8], we monitor patients in our department through regular consultations and examination with otoendoscopy and microscopy, and we only use diffusion-sequencing MRI in cases of suspected recurrence or complications, since diffusion-sequencing MRI has high reliability for the detection of cholesteatoma, especially when they are larger than 2 mm [14].

Similarly to other authors [15,16,17], we find that one of the fundamental contributions in cholesteatoma surgery is the use of endoscopes at 0 and 30°, which allows revision of the anterior epitympanum, tympanic sinus, and antrum, reducing the incidence of persistence and recurrence in these areas. 

As with other research [18], the attic was found to be the most common site of recurrence. In fact, all instances of recurrence were in this area for both techniques. On the other hand, most cases of disease persistence were found to be in the sinus tympani during the second-look revision (4 of 6 patients).

Furthermore, in other research [16], the surgery time was shorter in the AE-AE group (146 min) than in the CWU group (196 min). This results in more optimal occupation of operating rooms and less discomfort for the patient. 

In the group of patients treated with AE-AE using otoendoscopes, the most frequent treatment of the chain was partial reconstruction with cartilage on the stapes superstructure. We preferred autologous material to other types of materials, as we find, in line with other authors, that it provides functional benefits and allows us to reinforce the tympanic membrane, achieving acceptable functional results with post-surgical thresholds of 41 dB and a residual ABG of less than 20 dB, consistent with other studies [19].

In 18.2% of the patients, we performed total reconstruction with titanium prostheses, obtaining thresholds of 46 dB and a residual ABG of less than 20 dB in 28.6% of the patients. We found no statistically significant differences between partial reconstruction with cartilage over stapes and total reconstruction with titanium prosthesis when studying ABG, which is consistent with the literature [20]. 

There are reports [21] of high-frequency SNHL after surgery to treat COM with cholesteatoma, where SNHL exceeded 15 dB in 11% of cases compared with one patient (2%) in the AE-AE group and four patients (19%) in the CWU group in our investigation. In our study, all of these patients had previous high-frequency affectations. After surgery, their PTA values worsened with high frequencies (>20 dB in a frequency higher than 2 kHz). This suggests the occurrence of labyrinthitis before surgery, as presented in another report [22]. We believe that the differences in both techniques and the fall in high-frequency threshold are needed for two procedures in the CWU group. 

## 5. Conclusions

The AE-AE procedure utilizing an endoscope is a safe technique that significantly reduces the rates of cholesteatoma persistence and recurrence and shows good auditive results. It allows patients to have excellent quality of life, normalized bathing, and a reduction in surgical revisions when used in the treatment of attic acquired cholesteatoma. There is a need for controlled clinical trials to demonstrate the benefits of the AE-AE method compared with other techniques. 

## Figures and Tables

**Figure 1 jcm-12-00049-f001:**
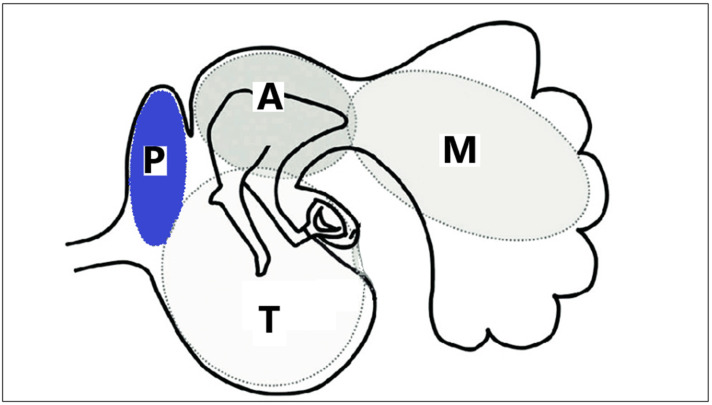
Schematic drawing of divisions of the tympanomastoid space. The tympanomastoid space is divided into four sections of the protympanum (P), the tympanic cavity (T), the attic (A), and the mastoid (M) in order to represent the extent of cholesteatoma (adapted from Tetsuya Tono et al., 2017 [1]).

**Figure 2 jcm-12-00049-f002:**
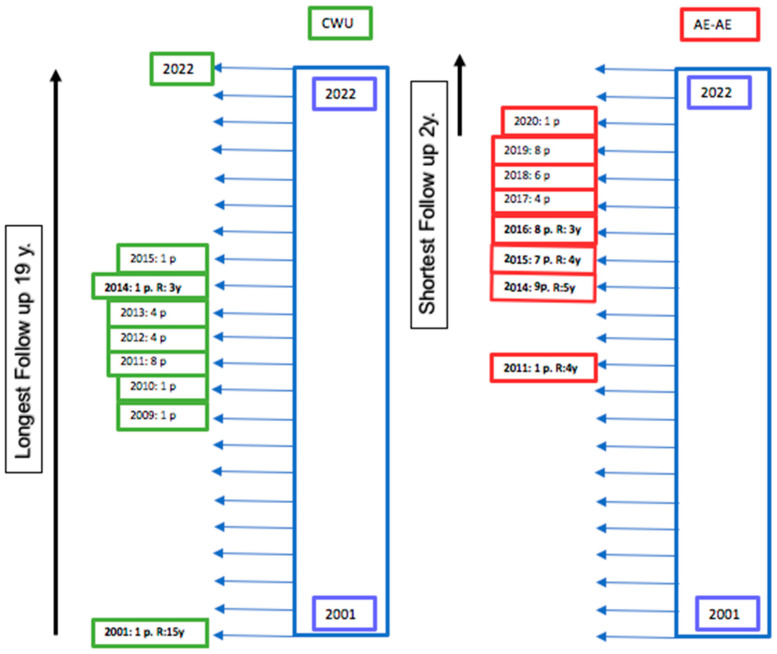
Scheme of patient inclusion, follow-up, and year of recurrence. CWU (in green) resulted in 2 patients with recurrence, and AE-AE (in red) resulted in 4 patients with recurrence. p: patients; R: recurrence.

**Figure 3 jcm-12-00049-f003:**
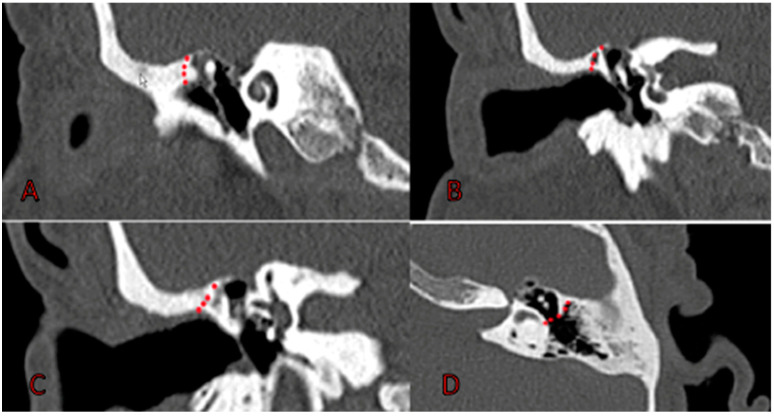
AE-AE hybrid exteriorization technique. (**A**–**C**) atticotomy starting anteriorly and progressing posteriorly without affecting the posterior wall or entering the antrum. (**D**) antrum exclusion with pieces of cartilage.

**Figure 4 jcm-12-00049-f004:**
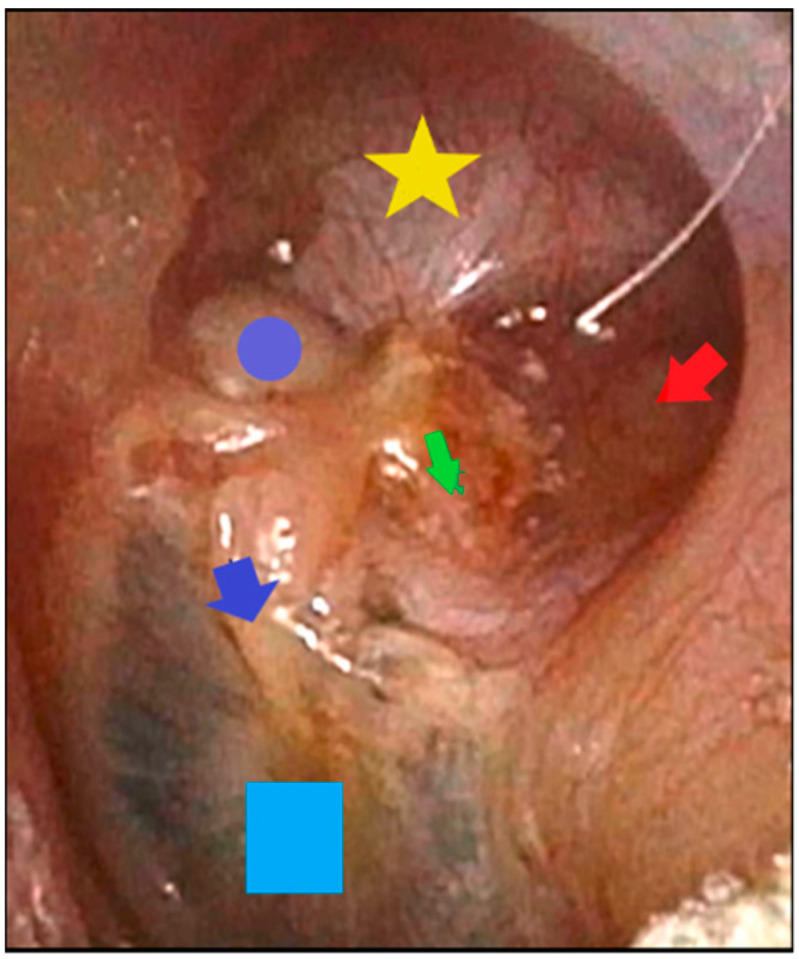
Post-surgical image of AE-AE without ossiculoplasty. Yellow star: atticotomy; blue circle: head of the malleus; blue arrow: handle of the malleus; green arrow: incus; red arrow: antroexclusion with cartilage; blue square: tympanic membrane.

**Figure 5 jcm-12-00049-f005:**
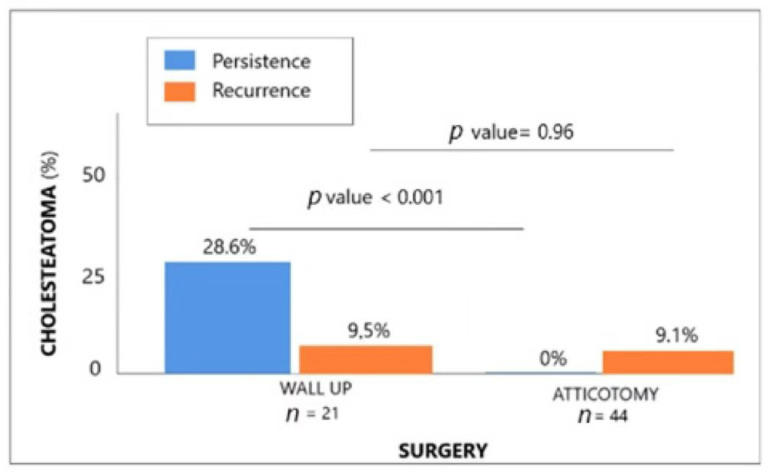
Comparison of recurrence and persistence of cholesteatoma rates according to type of surgery.

**Figure 6 jcm-12-00049-f006:**
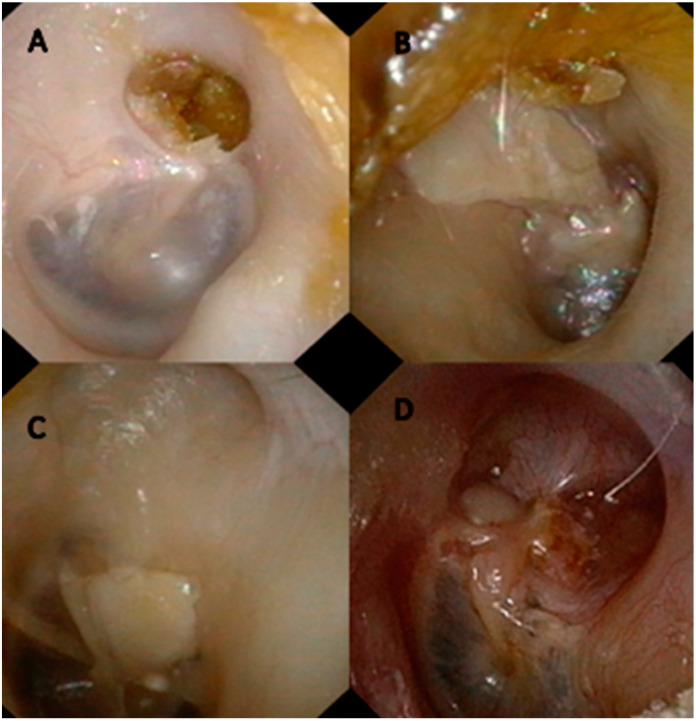
(**A**) Stage II attic cholesteatoma starting to affect the tympanic cavity (this is an ideal case for performing AE-AE). (**B**) Results of AE-AE procedure with cartilage over stapes. (**C**) Results of AE-AE procedure with cartilage over TORP. (**D**) Results of AE-AE procedure without ossiculoplasty.

**Table 1 jcm-12-00049-t001:** Japan Otological Society classification of cholesteatoma stages (adapted from Tetsuya Tono et al., 2017 [1]).

Stage
I: Cholesteatoma localized in the attic.
Ia: Retraction pocket with epithelial self-cleaning function.
Ib: Retraction pocket with persistent accumulation of keratin debris.
II: Cholesteatoma involving two or more sites.
III: Cholesteatoma with intratemporal complications or pathologic complications.
IV: Cholesteatoma with intracranial complications.

**Table 2 jcm-12-00049-t002:** Clinical, surgical, and recurrence characteristics according to type of surgery.

	Type of Surgery		*p*-Value
	Wall-Up *n* = 21	Atticotomy *n* = 44	
**Age** (years)	48 ± 20	43 ± 16	0.23
**Sex** (male) (*n* (%))	17 (81)	27 (61)	0.11
**Year of inclusion of patients** (*n* (%)) 2001 2009 2010 2011 2012 2013 2014 2015 2016 2017 2018 2019 2020	1 (4.8) 1 (4.8) 1 (4.8) 8 (38.0) 4 (19.0) 4 (19.0) 1 (4.8) 1 (4.8)	1 (2.3) 9 (20.5) 7 (15.9) 8 (18.1) 4 (9.1) 6 (13.6) 8 (15.9) 1 (2.3)	
**Affected ear** (right) (*n* (%))	10 (47.6)	25 (56.8)	0.49
**Surgery time** (min)	196 ± 57.1	146 ± 30.7	0.005
**Healing time** (days)	-	59 ± 36.7	
**Stage of cholesteatoma** (*n* (%)) Ib II	5 (24) 16 (76)	18 (41) 26 (59)	0.18
**Restoration of ossicular chain continuity** (*n* (%))	18 (86%)	34 (77%)	0.52
**Type of reconstruction of ossicular chain** (*n* (%)) No Stapes stabilizing cartilage graft Total prosthesis	3 (14.3) 11 (52.4) 7 (33.3)	10 (22.7) 26 (59.1) 8 (18.2)	0.36
**Facial nerve dehiscence** (*n* (%))	3 (15.8)	8 (15.9)	0.99
**Recurrence and persistence of cholesteatoma** (*n* (%)) Recurrence Residual cholesteatoma	2 (9.5) 6 (28.6)	4 (9.1) 0 (0)	0.96 0.0002
**Recurrence by stage (*n* = 6)** (*n*/N(%)) Stage Ib Stage II	(0) 2 (12.5)	1 (5.7) 3 (11.5)	0.99 0.99
**Persistence by stage (*n* = 6)** (*n*/N(%)) Stage Ib Stage II	0 (0) 6 (37.5)	0 (0) 0 (0)	0.99 0.002

Expressed as median (P_25_–P_75_).

**Table 3 jcm-12-00049-t003:** Audiological results comparing types of surgery and reconstruction.

	Type of Surgery		
	Wall-Up *n* = 21	Atticotomy *n* = 44	*p*-Value
**Audiometry** (Db)			
Global			
Pre-surgery	53 ± 16	44 ± 15	0.039
One year or more post-surgery	50 ± 17	42 ± 15	0.10
***p*-value**	0.66	0.95	
Comparing type of reconstruction			
None			
Pre-surgery	50 ± 20	37 ± 24	0.28
Post-surgery	48 ± 19	39 ± 18	0.50
***p*-value**	0.52	0.47	
Stapes stabilizing cartilage graft			
Pre-surgery	54 ± 16	45 ± 13	0.09
Post-surgery	50 ± 20	41 ± 14	0.20
***p*-value**	0.66	0.67	
Total prosthesis	45 ± 10	47 ± 10	0.69
Pre-surgery	57 ± 13	46 ± 18	0.35
***p*-value**	0.22	0.41	
**PTA *** (less than 20 Db) (*n* (%))			
Global	19 (43.2)	11 (52.4)	0.49

* Comparing one year or more audiometries with pre-surgery audiometries.

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
