# Peer review of "Effectiveness and Safety of Attic Exposition–Antrum Exclusion versus Canal Wall-Up in Patients with Acquired Stage Ib and II Cholesteatoma Affecting the Attic and Tympanic Cavity"

_jcm, 2022, doi:10.3390/jcm12010049_

Round 1

Reviewer 1 Report (New Reviewer)

Thank you for the possibility to review your paper. 

The topic is important. Main problems are the different situations of the two groups as you already mentioned.

I would suggest to refine the englisch text. Some sentences do not make sense in my opinion for instance:

"In this case we reviewed and historical cohort...."

1. What is the main question addressed by the research? Is the technique of Attic Exposition-Antrum Exclusion" as effective as canal wall-up? 2. Do you consider the topic original or relevant in the field? Does it address a specific gap in the field? In my opinion it is an interesting issue. 3. What does it add to the subject area compared with other published material? The described technique is an alternative to the more common surgical procedures and shows a similar safety. 4. What specific improvements should the authors consider regarding the methodology? What further controls should be considered? As it is a retrospective investigation it is difficult to change this methodology. 5. Are the conclusions consistent with the evidence and arguments presented Yes. and do they address the main question posed? Yes. 6. Are the references appropriate? Yes 7. Please include any additional comments on the tables and figures. The drawing is useful for the understanding of the procedure and the tables prove the results.

Author Response

In accordance with the reviewer's suggestion, we have submitted the manuscript for review by MDPI's Editorial Services.

Reviewer 2 Report (New Reviewer)

The topic is interesting and attempts to study the very vexing problems faced by otologists in cholesteatoma surgery. However, there are too many variables that could potentially influence the outcome in these 2 groups; the generalists vs the otologists, use of the endoscopes along with microscopes, follow up lengths are not clear.

Some sentences make no sense (possible language limitations). 

Duplicate tables seem to be included.

Author Response

Thank you for your comments, which have helped us to improve the manuscript.

Comments and Suggestions for Authors

The topic is interesting and attempts to study the very vexing problems faced by otologists in cholesteatoma surgery.

However, there are too many variables that could potentially influence the outcome in these 2 groups; the generalists vs the otologists, use of the endoscopes along with microscopes, follow up lengths are not clear.

There are a lot of variables that can affect the results, but the main variable is cholesteatoma A-T 1b-2 in the same population. We have a big homogeneous sample of this type of cholesteatoma but we can’t do the same with the surgeons because this is a retrospective study. Regardless this limitation we have a series of cases good enough to be studied. In the future we hope to do a randomized clinical trial but we think this study it’s a good star for our group and for other surgeons who want to try another option for this specific type of cholesteatoma

Some sentences make no sense (possible language limitations). 

In accordance with the reviewer's suggestion, we have submitted the manuscript for review by MDPI's Editorial Services.

Duplicate tables seem to be included.

We have deleted duplicate tables and improved the presentation.

Thank you again.

Reviewer 3 Report (New Reviewer)

This is a well written paper with excellent results that reinforces this newer approach to limited cholesteatoma. It supports the use of endoscopes in chronic ears by significantly reducing the recurrent disease and at the same time providing better audio metric outcomes ,shorter operative times, and the elimination of the second look procedure.The tables could be improved with better line adjustment of the categories and numbers.

Author Response

This is a well written paper with excellent results that reinforces this newer approach to limited cholesteatoma. It supports the use of endoscopes in chronic ears by significantly reducing the recurrent disease and at the same time providing better audio metric outcomes ,shorter operative times, and the elimination of the second look procedure.

Thank you for your comments

The tables could be improved with better line adjustment of the categories and numbers.

According to your comment, we have adjusted the categories and numbers in the tables.

We have submitted the manuscript for review by the MPDI Editing Service for extensive editing of English language and style.

Round 2

Reviewer 2 Report (New Reviewer)

Thank you for your response. The article appears much better now. There is still room for improvement with the language though.

This manuscript is a resubmission of an earlier submission. The following is a list of the peer review reports and author responses from that submission.

Round 1

Reviewer 1 Report

You describe the surgical results between AE-AE and canal wall-up in patients with acquired cholesteatoma stage IB and II. It is an interesting paper, however, It still needs considerable revision.

1. Please describe the surgeon which gives the impression of a high recurrence rate in canal wall-up.

2. Although not statistically significant, there is more stage 2 cases in the canal wall-up group; the recurrence rate per stage should also be described.

3. Please describe the operating time for each of them.

Author Response

To the reviewer:

Thank you for your comments as we believe they have sincerely helped to improve the manuscript. We provide a point-by-point response to the comments. Changes to the manuscript are highlighted in red.

Thank you again,

Dr. Francisco Arias-Marzan.

Reviewer 2 Report

The authors compared the clinical outcome of cholesteatoma surgery between EA-EV and canal wall-up mastoidectomy. During cholesteatoma surgery, complete removal of cholesteatoma is crucial for surgical outcomes. In this regard, an appropriate surgical field of view is critical.

1. The specific procedure of canal wall-up mastoidectomy was not described in the manuscript. During CWU mastoidectomy, the authors used oto-endoscopy. I think that otoendoscopy might be needed to detect cholesteatoma in sinus tympani. In addition, the recurrence rate of the CWU procedure was relatively high.

2. EA-EV surgery was performed under microscopy and ear-endoscopy. The concept of endoscopic assisted middle ear surgery has to be emphasized.

3. Indication of each surgical technique should be addressed in detail. Does a single surgeon perform the surgical procedures?

4. it is better to differentiate between recurrence and residual cholesteatoma as clinical outcomes.

Author Response

(The authors gave the same response as above.)

Round 2

Reviewer 1 Report

The manuscript is much improved.

Author Response

Responses to the Reviewer:

Thank you for your esteemed help in improving the manuscript.

Dr. Francisco Arias-Marzan.

Reviewer 2 Report

The authors have addressed issues.

However, I have questions about procedures of conventional surgery

1. The authors have described the surgical step of canal wall up mastoidectomy. I wonder whether post tympanotomy was performed in all cases. In addition, the authors described “complete mastoidectomy with sinus tympani exposure”, however, we could not see sinus tympani during canal wall up or down approach under microscopy.

2. This is the result of comparing the surgical performance of otologists(2 surgeons) and general otolaryngology specialists (Five surgeons). I wonder if the results would have been changed (better result) if the otologists had performed the CWU procedure. The authors must consider the issues regarding surgeon factor for cholesteatoma surgery.

Author Response

(The authors gave the same response as above.)
